# Early Measurement of Blood sST2 Is a Good Predictor of Death and Poor Outcomes in Patients Admitted for COVID-19 Infection

**DOI:** 10.3390/jcm10163534

**Published:** 2021-08-11

**Authors:** Marta Sánchez-Marteles, Jorge Rubio-Gracia, Natacha Peña-Fresneda, Vanesa Garcés-Horna, Borja Gracia-Tello, Luis Martínez-Lostao, Silvia Crespo-Aznárez, Juan Ignacio Pérez-Calvo, Ignacio Giménez-López

**Affiliations:** 1Department of Internal Medicine, Hospital Clínico Universitario, Lozano Blesa, 50009 Zaragoza, Spain; jorgerubiogracia@gmail.com (J.R.-G.); Vanesa_Garces@hotmail.com (V.G.-H.); bcgracia@salud.aragon.es (B.G.-T.); silviacrespoaz@gmail.com (S.C.-A.); jiperez@unizar.es (J.I.P.-C.); 2Aragon Health Research Institute (IIS Aragón), 50009 Zaragoza, Spain; npfrosyoly@gmail.com (N.P.-F.); lmartinezlos@salud.aragon.es (L.M.-L.); igimenez.iacs@aragon.es (I.G.-L.); 3Facultad de Medicina, University of Zaragoza, 50009 Zaragoza, Spain; 4Department of Immunology, Hospital Clínico Universitario, Lozano Blesa, 50009 Zaragoza, Spain; 5Instituto Aragonés de Ciencias de la Salud (IACS), 50009 Zaragoza, Spain

**Keywords:** soluble ST2, biomarker, COVID-19, prognosis, lung injury

## Abstract

Although several biomarkers have shown correlation to prognosis in COVID-19 patients, their clinical value is limited because of lack of specificity, suboptimal sensibility or poor dynamic behavior. We hypothesized that circulating soluble ST2 (sST2) could be associated to a worse outcome in COVID-19. In total, 152 patients admitted for confirmed COVID-19 were included in a prospective non-interventional, observational study. Blood samples were drawn at admission, 48–72 h later and at discharge. sST2 concentrations and routine blood laboratory were analyzed. Primary endpoints were admission at intensive care unit (ICU) and mortality. Median age was 57.5 years [Standard Deviation (SD: 12.8)], 60.4% males. 10% of patients (*n* = 15) were derived to ICU and/or died during admission. Median (IQR) sST2 serum concentration (ng/mL) rose to 53.1 (30.9) at admission, peaked at 48–72 h (79.5(64)) and returned to admission levels at discharge (44.9[36.7]). A concentration of sST2 above 58.9 ng/mL was identified patients progressing to ICU admission or death. Results remained significant after multivariable analysis. The area under the receiver operating characteristics curve (AUC) of sST2 for endpoints was 0.776 (*p* = 0.001). In patients admitted for COVID-19 infection, early measurement of sST2 was able to identify patients at risk of severe complications or death.

## 1. Introduction

Coronavirus disease 2019 (COVID-19) is a worldwide pandemic caused by the novel severe acute respiratory syndrome coronavirus 2 (SARS CoV-2), an infectious disease leading to high morbidity and mortality [1,2,3].

Clinical spectrum of COVID-19 ranges from asymptomatic patients to severe pneumoniae, with multi-organ symptoms and/or failure, including cardiological, neurological or thrombotic manifestations [4]. Prognosis depends on the development of an acute respiratory distress syndrome (ARDS), which frequently leads to the administration of high oxygen flow therapy (HOF) or admission at the intensive care unit (ICU) and mechanical ventilation, with a very high mortality in such cases [1,3].

As a novel disease, patient handling was initially based on the most conspicuous clinical signs and known biomarkers of tissue damage or inflammation such us lactate dehydrogenase (LDH), interleukin 6 (IL-6), ferritin, C-reactive protein (CRP) or lymphocytes [1,3]. However, these biomarkers lack specificity, since they can rise in other critical diseases [5], showing a poor prognostic value. For instance, the observation of heart damage and myocarditis associated to COVID-19 led to investigation of troponin, whose concentrations have been found to correlate poorly with COVID-19 prognosis [6,7,8]. Accordingly, the search for more specific, earlier and clinical useful biomarkers of poor outcomes remains crucial.

Derangement of immune response was one of the first observations in COVID-19 and still remains as a hallmark of progression into a more severe phase. Studies of lung immune cells infiltrates suggest a role for IL-33/ST2 axis [9].

Interleukin-1 receptor-like-1 (IL1RL1), also known as suppression of tumorigenicity 2 (ST2), is a member of interleukin (IL)-1 receptor family, with two main isoforms, a transmembrane cellular (ST2L) and a circulating soluble form (sST2). ST2 is the receptor for IL-33, a cytokine released by cells in response to cell damage or stress (‘alarmin’). IL-33 binding to ST2L elicits a pleiotropic action. It is well stablished that IL-33/ST2 axis exhibits a cardioprotective role, reducing fibrosis, cardiomyocyte hypertrophy and apoptosis and improving myocardial function [10]. For that reason, blood sST2 levels are clinically used to assess prognosis in heart failure (HF). It is assumed blood sST2 reflects tissue sST2 acting as a decoy receptor for IL-33 and thus reducing IL-33 cardioprotective effects. ST2 has also been associated with inflammatory phenomena [11], especially on the vascular endothelium, and pulmonary tissues [9,10].

The study by Pascual-Figal et al. demonstrated the lung origin of sST2 in acute decompensated HF, finding a correlation between their concentrations and alveolar wall thickness. They also found sST2 is higher in patients with non-cardiogenic pulmonary edema [9,12]. Moreover, there is also convincing evidence for a role of sST2 in lung diseases such as asthma, Chronic Obstructive Pulmonary Disease (COPD) and ARDS [13,14].

We hypothesized that sST2 circulating levels in COVID-19 patients reflects pulmonary damage and the intensity of inflammatory response elicited by SARS-CoV2, and thus could be a clinical useful biomarker. Supporting that hypothesis, we concluded that in patients admitted for COVID-19 infection, early measurement of sST2 was able to identify patients at risk of severe complications or death.

## 2. Materials and Methods

### 2.1. Study Design and Setting

A prospective cohort study was carried out at the Infectious diseases and Internal Medicine departments of a tertiary teaching center (Hospital Clínico Universitario “Lozano Blesa”, Zaragoza, Spain), between July and November 2020. Inclusion criteria: (1) Age ≥ 18 years. (2) Informed consent granted. (3) Confirmed diagnosis of SARS-CoV2 (COVID-19) infection by nasopharyngeal polymerase chain reaction (PCR) or specific serology (IgM and/or IgG) in the context of clinical respiratory infection. Exclusion criteria: (1) Primary admission at Intensive Care Unit. (2) Refusal to participate. (3) Functional dependence (Barthel index < 50 points). (4) Moderate/Severe cognitive impairment (Pfeiffer scale ≥ 5). (5) Advanced chronic obstructive pulmonary disease (COPD) (FEV1 < 30%) or a history of emphysema and/or pulmonary fibrosis. (6) Active cancer.

The study complied with the fundamental guidelines of the Helsinki declaration guidelines and was evaluated and approved by Aragón’s Committee on Research Ethics (CEICA, Ref. PI20/248, 13 May 2020).

### 2.2. Variables and Definitions

Patients were studied at three specific times during COVID-19 hospitalization. (1) ‘Admission’ (first 24 h upon admission), (2) ‘Control’ (48–72 h later) and (3) ‘Discharge’ (last 24 h prior to discharge). At each time point, vital signs were recorded (blood pressure, heart rate, oxygen saturation, and respiratory therapy), Kirby index (PaO_2_/FiO_2_) was estimated (ePAFI) from FiO_2_ and oxygen saturation, and patient’s dyspnea was quantified using Borg scale (range 1 [minimum] to 10 [maximum]). Routine blood laboratory data (Complete blood Countbiochemistry, coagulation, and arterial blood gasses) were recorded.

Primary outcomes were death and ICU admission. Secondary outcomes were length of stay, need for HOF, increase treatment or both of them at 48/72 h. Deaths occurring after ICU admission were not accounted for.

### 2.3. Circulating sST2 Measurements

Serum sST2 concentrations were determined in 150 COVID-19 patients. Blood samples were withdrawn at admission, control and discharge. Blood was collected into clotting gel test tubes, centrifuged and serum was aliquoted and stored (Aragón’s Health System Biobank) at −80 °C until analysis. Eventually, 144 admission and control, and 80 discharge samples were processed and analyzed. Serum aliquots were virus-inactivated by treatment with 1% Triton-X100. On the day of the analysis, serum was thawed and diluted 1:50 (1% Bovine Serum Albumin in Phosphate Buffered Saline). Serum concentrations of soluble ST2 were determined by sandwich enzyme-linked immuno-sorbent assay (ELISA), following manufacturer’s instructions. (Appendix A). A set of sera from 60 healthy donors obtained through Aragón’s Health System Biobank (BSSA) was also analyzed in this manner. These sera had been originally collected from two independent sources, and were selected to match patient cohort age and gender distribution. Random samples from COVID patients and healthy donors were re-run in independent assays to test and correct for inter-assay variability.

### 2.4. Statistical Analysis

Continuous variables were expressed as mean ± standard deviation (SD) or median (Inter Quartile Range, IQR), as appropriate depending on normality. Categorical variables were expressed as percentages. To perform the comparative analysis between normal continuous variables, ANOVA test was used. Those variables not following normality were compared using a Kruskal–Wallis U test. Categorical variables were compared using the chi-square test. The analysis of the different correlations between continuous variables was carried out using the Pearson or Spearman test for continuous variables and the Mantel–Haenszel test for ordinal categorical variables.

A goal of 150 inclusions was set to account for a 20% of losses, which were mostly due to the assistance pressure imposed on the clinician researchers by the pandemic situation.

In univariable and multivariable logistic regression analysis, sST2 concentration was dichotomized based on a cut-off value selected from receiver operating characteristic (ROC) curves analysis of its primary endpoint predictive value. Multivariable logistic regression model was designed to identify factors independently associated with the need of ICU transfer during admission or intra-hospital death. Candidate predictors were selected from the univariable analysis when *p*-value < 0.200 and entered using a backward selection procedure. Age, gender and previous history of diabetes as described clinical risk factors, were also included in the model. Continuous candidate variables were log-transformed (lnX) if necessary.

Confidence intervals included were 95% (95% CI), establishing statistical significance with a *p* lower than 0.05. Statistical analysis was carried out with Statistical Package for the Social Sciences (SPSS) version 24.0 for Windows (IBM).

## 3. Results

### 3.1. Hospitalized COVID-19 Patients Exhibit Elevated Serum sST2 Concentrations

A total of 150 patients were sequentially included (inclusion Flow-chart depicted in Appendix A). Participant’s mean age was 57.5 ± 12.8 years. In total, 60% were male and 50% had bilateral pneumoniae. Serum concentrations of sST2 were determined in COVID-19 patients and age- and gender-matched healthy donors.

Median serum sST2 values from 60 healthy donors were 18.6 (15) ng/mL. We found no differences across age groups (Appendix A), but sST2 concentrations were significantly higher in female donors (female 22.3[13] vs. male 15.7[11] ng/mL, *p* = 0.024); Appendix A.

A total of 369 COVID-19 blood samples were collected and analyzed for sST2 concentrations (144 at admission, 145 at control time and 80 at discharge). Admission serum sST2 levels were significantly higher than in healthy donors (*p* < 0.001); Appendix A. Serum sST2 levels at admission were higher in males (males 59.6(38) ng/mL vs. females 45.4(26); (*p* = 0.012); Appendix A). There were no differences in sST2 concentrations across age groups at any sampling time (Appendix A).

In COVID-19 patients sST2 concentrations peaked at 48–72 h control measurement, representing on average a 150% increase over admission (53.1 ng/mL (IQR: 30.9) admission vs. 79.3 (IQR: 64.2) control; *p* < 0.001). sST2 concentrations at discharge had significantly declined (44.9 ng/mL (IQR: 39.6); *p* < 0.001) but were still significantly above healthy donor’s levels; Figure 1.

### 3.2. Serum sST2 Levels Correlate with Clinical and Laboratory Index of Disease Severity

For the initial analysis of clinical meaning for the increase in serum sST2, patients were divided into three groups according to their values for serum sST2 at admission. Table 1 summarizes baseline characteristics in the whole cohort and stratified by sST2 terciles.

As shown in Table 1, patients in the upper tercile for sST2 at admission (>percentile 75; cut-off = 70.4 ng/mL), tended to be male in a higher proportion (*p* = 0.066), and showed no differences for comorbidities.

Patients in this *p* > 75 group had worse respiratory function as assessed by estimated PAFI (*p* < 0.001), higher concentrations of aspartate aminotransferase (AST) (*p* < 0.001), alanine aminotransferase (ALT) (*p* = 0.006), creatin–phosphokinase (CK) (*p* = 0.007), lactate dehydrogenase (LDH) (*p* ≤ 0.001), C-reactive protein (*p* = 0.005), ferritin (*p* = 0.007), D-Dimer (*p* = 0.030), fibrinogen (*p* = 0.011) and interleukine-6 (*p* = 0.008) and higher total lymphocyte count (*p* = 0.021). There were no differences in treatment schedule at admission.

Admission sST2 concentrations showed a significant negative correlation with estimated PAFI (r = −0.361; *p* ≤ 0.001), and significant positive correlations with LDH (r = 0.328; *p* < 0.001), C-reactive protein (r = 0.274; *p* = 0.001) and IL-6 serum concentrations (r = 0.271; *p* ≤ 0.001).

### 3.3. Serum sST2 Concentrations Associate with Adverse Outcomes

In total, 15 out of 144 (10.4%) patients reached primary endpoint (clinical characterization based in outcome incidence is shown as Appendix A). A total of 14 patients required admission to ICU. One patient died of bacteriemia associated to a central venous catheter. Patients with the highest sST2 admission concentrations (>percentile 75), experienced the primary outcome in a higher proportion (P75 = 25.7% vs. P25 = 0%; *p* = 0.001, Table 2). A similar result is observed if the comparison is made based on sensitivity threshold (Appendix A).

Admission sST2 showed the best area under the curve (AUC = 0.776; *p* = 0.001) compared with LDH, IL-6, ferritin, CK and C-reactive protein (Figure 2). A cut-off value of 58.9 ng/mL identifies the primary outcome with the best Sensitivity (78.6%) and Specificity (60.8%). Based on this cutoff, Kaplan Meier survival curves for the primary endpoint by sST2 concentrations at admission were generated and showed significantly differences in mortality and ICU admissions (Log-rank test ≤ 0.001; Figure 3).

Median length of stay for patients not requiring admission to ICU was 8 days (IQR 6). sST2 values at admission or control time points, or the increase between them, did not correlate with length of hospital stay. However, there was a negative correlation between length of hospital stay and sST2 at discharge (Spearman’s rho = −0.338, *p* = 0.003).

At control time, 47 patients (34.1%) needed HOF, oxygen administration had to be increased in 53 patients (37.9%), and 66 patients (48.5%) required intensification of treatment (Table 2). Proportion of patients suffering these secondary outcomes was higher in the > P75 admission sST2 group for all three events, but only reached significance for HOF (>percentile 75 = 51.5% vs. P25 = 28.6%; *p* = 0.053).

Univariable logistic regression analysis identified ePAFI (HR 0.98 [0.97–0.99]; *p* = 0.001), CK (HR 2.06 [1.01–4.20]; *p* = 0.047) and admission sST2 > 58.9 ng/mL (HR 6.32 (1.70–23.5); *p* = 0.006), as potential predictors for primary endpoint.

In the multivariable logistic regression model, after adjusting for confounders, including age, gender, body mass index (BMI), previous history of diabetes and previous history of dyslipidemia, admission sST2 concentrations (HR 9.73 (2.12–44.8); *p* = 0.030) was identified as independent predictor for the primary endpoint (Table 3). AUC of the multivariable model improved significantly when sST2 admission concentrations were included (0.71 (0.567–0.871) vs. 0.81 (0.70–0.92)) (Table 3).

## 4. Discussion

Here we present our results demonstrating that early changes in blood sST2 have an excellent prognostic value in COVID-19, surpassing more traditional markers having been used during the pandemic. An added value to our findings is the fact they have been produced from a cohort of patients managed under usual clinical practice conditions.

The emergence of COVID-19 has posed an unprecedent challenge for clinicians around the world. COVID-19 exhibits a wide clinical spectrum ranging from asymptomatic or mild respiratory tract symptoms to the development of ARDS and death. Respiratory failure is usually the hallmark of bad prognosis and mortality [1,2]. The reasons why patients, presenting with apparently similar clinical features, evolve to a severe disease or keep mildly affected remain to be fully elucidated. Even worse, the development of severe lung affectation has been proven difficult to predict. Moreover, COVID-19 is putting significant strain on healthcare systems, and tools assisting in evidence-guided decision making to allocate limited resources are still in urgent need.

Several clinical scores have been developed or applied in an attempt to predict COVID-19 prognosis. Such scores imply systematic and quantitative assessment of clinical features and traditional, readily available laboratory tests [15,16,17,18,19]. Given the complex physiopathology involved in COVID-19 the disease has been considered for AI and machine learning based approaches for prognosis prediction [20,21]. Among the biomarkers most frequently added in predictive models, there are classic indicators of cell damage (LDH, troponin) and biomolecules suspected to be involved in the immune and inflammatory response firstly identified in disease pathogenesis (IL-6, ferritin, or lymphocytes count) [1,3,22]. A recent meta-analysis, including results from 32 studies, mostly focused on China and USA, found the most valuable markers were decreased lymphocyte count, a decreased platelet count and elevated C reactive protein, creatine kinase, procalcitonin, D-dimer, lactate dehydrogenase, alanine aminotransferase, aspartate aminotransferase, and creatinine [23]. None of them are specific, though, complicating risk stratification in a disease following worse course in patients with chronic comorbidities characterized by low-degree inflammation.

Moreover, in most regular clinical settings, and under time and resources constraints experienced during infection waves, more simple and straightforward methods are yet convenient. We have previously reported that a simple point-of-care lung ultrasound evaluation is able to predict worse outcome in hospitalized COVID-19 patients [24]. Now, seeking for a potential involvement in COVID-19 of biomarkers traditionally associated to lung deterioration during acute heart failure decompensation (ADHF), we found that admission sST2 values correlated well with biochemical markers and indexes being used to evaluate clinical course in COVID-19. Indeed, multivariate regression models and survival analysis demonstrated that admission sST2 is an independent predictor of worse prognosis in patients admitted for COVID-19. In ROC analysis sensitivity and specificity of sST2 to predict worse outcomes was significantly higher than that of the aforementioned biomarkers. sST2 correlated positively to prescription of HOF, transfer to ICU and death. Altogether, our findings support sST2 clinical usefulness as individual prognostic marker in hospitalized COVID-19 patients.

Since we reported our results in preprinted form [25] there have been several studies determining sST2 blood levels in COVID-19 patients. As the IL-33/ST2 axis is involved in immune response to viral infections, Zeng et al. measured IL-33 and sST2 in 80 COVID-19 patients [26]. In line with our results, sST2 levels were significantly elevated in COVID-19 and correlated with other markers of inflammation and disease clinical severity. However, this study did not address hard clinical endpoints like ICU admission or death [26]. Interestingly, this study did not find significant differences in serum IL-33 between healthy controls and COVID-19 patients. However, a significant association for IL-33 levels at admission with adverse outcome (ICU admission, requiring ventilation or death) has been reported, with a very good predictive value (AUROC 0.83) in patients < 70 years [27]. Another study looking for cytokine signature and prognosis association in 175 COVID-19 patients found that sST2 was, together with seven other markers, independently associated with mortality both at baseline and longitudinally. On the other hand, IL-33 increased moderately in critically ill COVID-19 patients but did not change in deceased ones [28]. Serum sST2 also associated to worse outcomes (severity and death) in a cohort of 100 patients hospitalized for COVID-19 being evaluated for endothelial activation and stress [29]. Our study also analyzed secondary outcomes such as need for additional therapy or oxygen/ventilatory support. Although there were associations between sST2 serum concentrations and need for HOF at 48/72 h, it did not reach statistical significance (*p* = 0.053). However, others have found that plasma sST2 can provide dynamic information about the degree of a patient’s lung injury and thus can be used to monitor ventilator dependency in ARDS caused by COVID-19 [30]. In a preprint report, Huang examined the plasma protein profile of COVID-19 patients and the overlaps with common comorbidities. Clinical evolution was estimated from longitudinal analysis of WHO 6-point ordinal scale. Baseline sST2 levels were associated with baseline disease severity and with worse outcomes (death or requiring ventilation in 28 days follow-up). sST2 values peaked at day 3 [31]. Thus, mounting evidence supports our finding that sST2 is elevated in COVID-19 and that such increase bears prognostic significance.

A unique aspect of our study is the demonstration that early changes predict outcome better than peak values, stressing their clinical usefulness in prognosis. Moreover, sST2 showed a decline in patients at discharge, displaying a fully dynamic behavior that is not present with other biomarkers. Interestingly, discharge sST2 correlated negatively with hospital stay length. This is suggestive that sST2 decline is delayed with regard to clinical improvement. Other reports have shown persistently elevated sST2 levels in severe COVID-19 cases as long as 30 days after clinical onset [26,28]. It remains to be elucidated whether sST2 will recover completely to normal values and whether persistent sST2 levels would associate to clinical sequalae in COVID-19 patients. The course of temporal changes in circulating sST2 might also be informative about disease pathophysiology.

The IL-33/ST2 axis might play a leading role in COVID-19 pathogenesis [32,33,34,35]. Although our study was not aimed to gain pathophysiological insight into COVID-19, our findings support this hypothesis. Based on data from early studies of local lung inflammation, cytokine/immune cell profiling and systemic responses, Zizzo et al. first hypothesized IL-33 has a pivotal role in immune and inflammatory response to SARS-CoV2 infection. The release of IL-33 alarmin by pneumocytes and endothelium, in response to cell damage caused by viral infection, triggers a cascade of events in fibroblasts, alveolar membrane, coagulation and immune system [32]. IL-33 effects would be auto-amplified by IL-33 induction of IL-33 release and ST2 expression and sST2 release, as observed recently for asthma [36]. IL-33 is a well-known factor in asthma, one of the several lung diseases where IL-33 plays a relevant role [36]. More importantly, IL-33 has also been shown to participate in lung epithelium response to viral infection [37] and to decrease antiviral innate immunity in this organ [38]. IL-33 has been shown to be upregulated and released following SARS-CoV-2 infection in human epithelial cells [33]. Due to technical advantages over IL-33 measurements, circulating sST2 has often been employed as a surrogate marker of IL-33/ST2 axis status. Several studies have shown serum sST2 concentrations are elevated, and correlate with prognosis, in patients with pulmonary disorders including asthma, idiopathic pulmonary fibrosis, severe sepsis and trauma [13]. Bajwa et al. compared sST2 concentrations in patients with ARDS vs. congestive HF. They found higher sST2 concentrations in ARDS, exhibiting good correlation with mortality and APACHE III scale; to conclude, sST2 could be a very good biomarker to distinguish ARDS from cardiogenic edema [14]. Of note, serum sST2 concentrations we determined in COVID-19 patients experimenting adverse outcomes were several folds higher than those we routinely measure in most severe acute decompensated heart Failure (ADHF) cases, using the same test.

Several organs and tissues other than the lung and immune cells, namely the heart, adipose tissue and endothelium, express locally the IL-33/ST2 axis and may release sST2, as a consequence of direct infection by SARS-CoV-2 or secondary to immune system activation [9,10]. The IL-33/ST2 axis is likely playing a role in COVID-19 systemic manifestations [32]. For instance, a rare, but severe pediatric manifestation in COVID-19 has been described which closely resembles Kawasaki disease [39]. sST2 has been shown to be elevated in patients with Kawasaki disease which, in addition, correlated with clinical worse course [40,41]. More importantly, cardiovascular co-morbidity, and more specifically HF, has been identified as a significant risk factor for COVID-19 severity and worse outcome [34,35,42]. sST2 is also an established prognosis maker in HF. sST2 blood concentrations correlate with mortality, alone or in combination with other classical biomarkers such as natriuretic peptides [43]. Interestingly, the study by Pascual-Figal et al. provided strong evidence that vascular and pulmonary endothelium, as well as pneumocytes are main sources of sST2 in ADHF [9,10]. Whether the observed elevated sST2 levels in COVID-19 bears any association with heart co-morbidity in these patients deserves further consideration, as does the possible consequences for the cardiovascular system of a potentially sustained elevation in sST2 in convalescent patients, identified in this and other studies [34].

Our study has some limitations. It has been conducted in a single center and on a small sample size, this latter driven by the urgency to advance our knowledge on the disease. Nonetheless, clinical characteristics in our cohort were concordant with published data from other studies around the world and in our country [1,2]. Due to resource limitations imposed by the demanding sanitary situation, access to imaging and functional tests was difficult to standardize for even a representative fraction of study patients and thus their clinical value could not be compared directly to that of sST2. Clinical data for the study were collected under conditions of real practice. Obviously, this may imply unforeseen bias have been introduced, but multivariable analysis diminished this risk. And in the other hand, in a such heterogenous syndrome and with pandemic assistance, this setting could be more a strength than a limitation.

Identification of sST2 as an early predictor of worst prognosis in COVID-19 patients has important clinical implications. sST2 is a readily available biomarker, which can be consistently determined in blood samples. Accumulating evidence supports that sST2 levels elevate early in SARS-CoV2 infection and that such changes represent an independent risk factor for worse outcome, outpowering other biomarkers that so far have been guiding clinical and therapeutical handling. Another consistent report is the persistence of elevated sST2 after apparent clinical resolution. Given the well-established associations of sST2 levels with other lung and cardiovascular diseases, it is mandatory to investigate whether such sST2 levels reflect some inflammatory memory with long-term consequences. Our findings also lend support to the hypothesis of an IL-33/ST2 axis led pathophysiology in COVID-19. Pharmacological modulators for this axis are available and currently being trialed for other pathologies, that could be evaluated for treating or preventing severe cases of COVID-19.

Among patients suffering from COVID-19 infection, severely enough to be admitted, measurement of sST2 within 24 h of admission may be a useful biomarker for an early identification of those at higher risk of severe complications or death and for implementing more aggressive therapies since the initial stages of the disease.

## Figures and Tables

**Figure 1 jcm-10-03534-f001:**
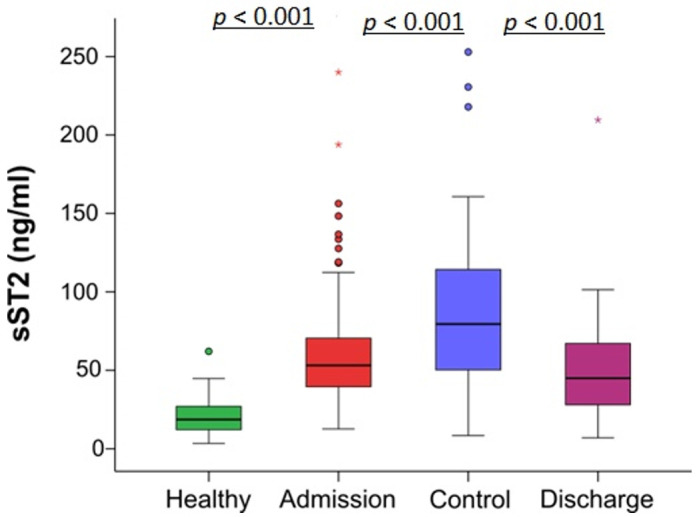
sST2 concentrations in healthy donors and COVID-19 patients. Box plot showing median, 1st and 3rd quartiles (box), 1.5 times values (whiskers) and outliers (dots) or extreme outliers (stars). Statistical significances were calculated using the Wilcoxon test.

**Figure 2 jcm-10-03534-f002:**
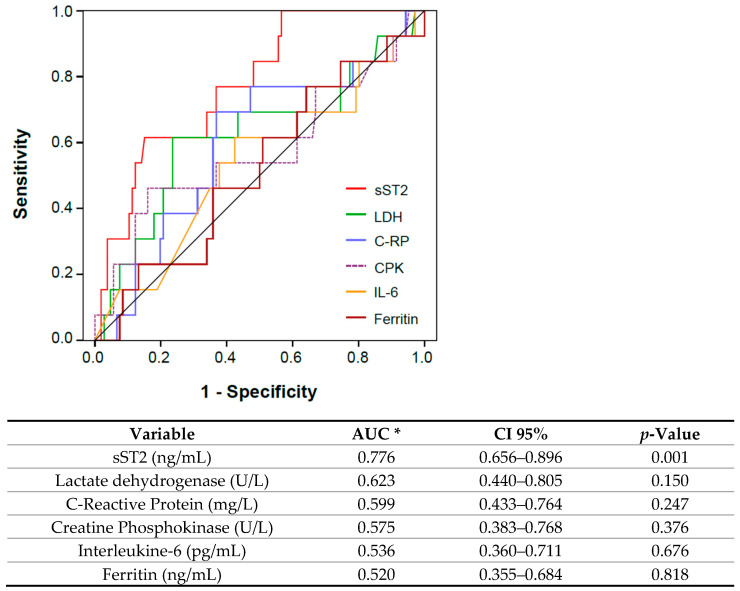
Receiver operating characteristic curves comparing sST2 concentrations at admission with C-reactive protein (C-RP), lactate dehydrogenase (LDH), ferritin and interleukine-6 (IL-6) concentrations at admission. * Area Under Curve.

**Figure 3 jcm-10-03534-f003:**
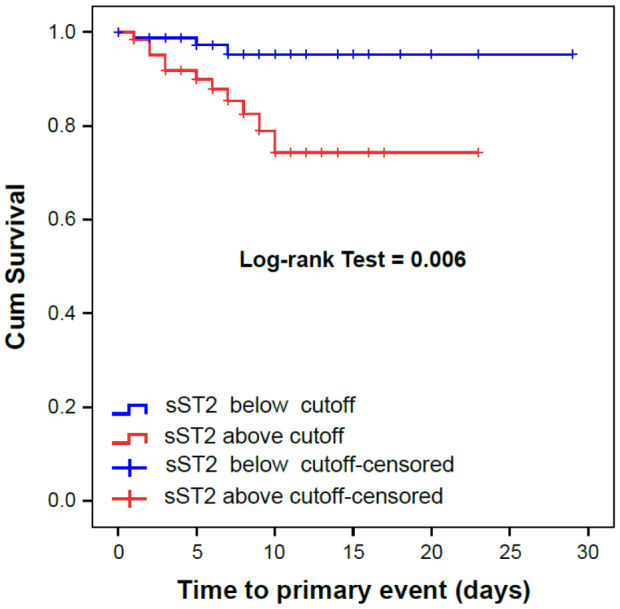
Kaplan Meier curves for the primary outcome (ICU admission for mechanical intubation and/or in-hospital death) by sST2 concentrations at admission.

**Table 1 jcm-10-03534-t001:** Baseline characteristics according to sST2 concentrations (terciles) at admission.

Variable	Total	*p* < 25	P25 to 75	*p* > 75	*p*-Value
Total size (*n*)	144	36	73	35	
Age (years)	57.5 ± 12.8	58.5 ± 12.3	57.2 ± 13.6	57.1 ± 12.0	0.878
Gender-Male (*n*(%))	87 (60.4)	17 (47.2)	44 (60.3)	26 (74.3)	0.066
Duration of symptom (days)	6.5 ± 3.3	7.0 ± 3.6	6.3 ± 3.3	6.3 ± 2.7	0.486
Time until COVID confirmation (Days)	3 (7)	3 (7)	3 (6)	3 (6)	0.492
Comorbidities (*n*(%)):					
- Hypertension	54 (37.5)	15 (41.7)	26 (35.6)	13 (37.1)	0.827
- Heart failure	4 (2.8)	1 (2.8)	2 (2.8)	1 (2.9)	1.000
- Dyslipidemia	42 (29.2)	9 (25.0)	19 (26.0)	14 (40.0)	0.267
- Coronary artery disease	5 (3.5)	1 (2.8)	3 (4.1)	1 (2.9)	0.914
- Diabetes	25 (17.4)	6 (16.7)	9 (12.3)	10 (28.6)	0.113
- History of smoking	48 (33.6)	10 (27.8)	22 (30.6)	16 (45.7)	0.207
- COPD/Asthma	16 (11.1)	3 (8.3)	10 (13.7)	3 (8.6)	0.605
- Atrial/flutter fibrillation	5 (3.6)	1 (2.9)	3 (4.3)	1 (2.9)	0.901
- CKD	7 (4.9)	3 (8.3)	2 (2.7)	2 (5.7)	0.427
Clinical variables					
- BMI (Kgs/m^2^)	28.9 (6.4)	27.5 (0.5)	28.7 (5.7)	30.0 (6.0)	0.434
- SBP (mmHg)	126.9 ± 16.7	132.5 ± 14.4	125.2 ± 15.7	124.6 ± 17.0	0.066
- DBP (mmHg)	77.2 ± 10.9	80.7 ± 11.6	77.0 ± 10.3	74.2 ± 10.6	0.051
- HR (bpm)	80.9 ± 12.8	80.9 ± 12.1	81.0 ± 12.8	80.5 ± 13.7	0.980
- Estimated PAFI (mmHg)	367 (92)	429 (101)	403 (99)	341 (108)	**<0.001**
- Borg scale for dyspnea (points)	4 (6)	3 (6)	5 (5)	4 (6)	0.486
Laboratory:					
- Urea (mg/dL)	33 (19)	38 (16)	32 (18)	32 (20)	0.069
- Creatinine (mg/dL)	0.94 (0.29)	0.91 (0.27)	0.88 (0.29)	0.92 (0.36)	0.318
Laboratory:					
- Aspartate aminotransferase (U/L)	37 (27)	30 (16)	38 (31)	41 (24)	**<0.001**
- Alanine aminotransferase (U/L)	31 (28)	23 (23)	32 (43)	33 (18)	**0.006**
- Creatin phosphokinase (U/L)	94 (92)	71 (80)	98 (92)	116 (143)	**0.007**
- Lactate dehydrogenase (U/L)	306 (145)	267 (70)	310 (106)	398 (208)	**<0.001**
- C-Reactive Protein (mg/L)	63 (81)	46 (63)	59 (68)	112 (137)	**0.005**
- Ferritin (ng/mL)	707 (908)	619 (838)	676 (813)	1338 (1061)	**0.007**
- Hemoglobin (g/dL)	14.2 ± 1.5	14.2 ± 1.3	14.3 ± 1.6	14.1 ± 1.4	0.234
- Total leucocytes (×1000)	5.6 (3.1)	5.2 (2.1)	5.4 (3.4)	6.1 (3.4)	0.251
- Total lymphocytes (×1000)	0.9 (0.7)	1.0 (0.5)	0.9 (0.6)	0.6 (0.6)	**0.021**
- D-Dimer (ng/mL)	688 (633)	719 (856)	625 (502)	976 (830)	**0.030**
- Fibrinogen (mg/dL)	775 (208)	739 (257)	761 (200)	811 (252)	**0.011**
- Interleukine-6 (pg/mL)	40 (30)	26.8 (32.4)	42.3 (26.2)	50.0 (24.3)	**0.008**
Chest X-Ray (*n*(%)):					0.222
- Normal	25 (17.9)	8 (23.5)	13 (18.3)	4 (11.4)	
- Unilateral pneumoniae	35 (25.0)	9 (26.5)	13 (18.3)	13 (37.1)	
- Bilateral pneumoniae	80 (57.1)	17 (50.0)	45 (63.4)	18 (51.4)	
Baseline therapies (*n*(%))					
- Colchicine	10 (6.9)	2 (5.6)	5 (6.8)	3 (8.6)	0.882
- Plasma	1 (0.7)	0 (0.0)	1 (1.4)	0 (0.0)	0.613
- Remdesivir	46 (31.9)	9 (25.0)	23 (31.5)	14 (40.0)	0.397
- Systemic corticosteroids	113 (78.5)	26 (72.2)	56 (76.8)	31 (88.6)	0.214
- Medium dose of corticosteroids (Dexamethasone (mg))	6 (3)	6 (3)	6 (3)	6 (3)	1.000
- Low molecular weight heparin	138 (95.8)	34 (94.4)	70 (95.9)	34 (97.2)	0.488

Variables are expressed as mean ± standard deviation or median (IQR). BMI: Body Mass Index; CKD: Chronic Kidney Disease (estimated glomerular filtration rate < 60 mL/min/173 m^2^ CKD-EPI—Creatinine method); COPD: Chronic Obstructive Pulmonary Disease; DBP: Diastolic Blood Pressure; HR: Heart Rate; bpm: beats per minute; SBP: Systolic Blood Pressure. **Bold results** are statistically significant.

**Table 2 jcm-10-03534-t002:** Outcomes by sST2 concentrations at admission.

Variable	Total	sST2 < P25	sST2 P25–P75	sST2 > P75	*p*-Value
Primary outcome (*n*[%]):					
• ICU admission and/or death	15 (10.4)	0 (0)	6 (8.2)	9 (25.7)	**<0.001**
Secondary outcomes:					
• Length of stay (days)	8 (6)	8 (6)	7 (5)	8 (7)	0.328
• Need for HOF at 48/72 h (*n*[%])	47 (34.1)	10 (28.6)	20 (28.6)	17 (51.5)	0.053
• Need to increase COVID-19 treatment at 48/72 h (*n*[%])	53 (37.9)	11 (30.6)	25 (35.7)	17 (50.0)	0.214
• Necessity of HOF or increase COVID-19 treatment at 48/72 h (*n*[%])	66 (48.5)	14 (40.0)	32 (47.1)	20 (60.6)	0.223

Variables are expressed as mean ± standard deviation or median (IQR). ICU: Intensive Care Unit. HOF: Higher O_2_ Flow Therapy. Statistical analysis of these variables was made with a Mantel–Haenszel test. **Bold results** are statistically significant.

**Table 3 jcm-10-03534-t003:** Univariable and multivariable logistic regression model for the primary combined endpoint all-cause mortality and/or ICU admissions.

Univariable			Multivariable	
Variable	HR (CI 95%)	*p*-Value	HR (CI 95%)	*p*-Value
Age	1.04 (0.99–1.09)	0.073		
Gender-male	1.38 (0.47–4.05)	0.555		
BMI	1.08 (0.98–1.16)	0.089		
Diabetes	2.73 (0.84–8.82)	0.094		
Dyslipidemia	3.19 (1.08–9.47)	**0.036**		
SBP	1.00 (0.97–1.03)	0.937		
DBP	0.97 (0.93–1.03)	0.366		
Estimated PAFI *	0.98 (0.97–0.99)	**0.001**		
Urea *	1.45 (0.55–3.83)	0.455		
Aspartate transaminase *	1.31 (0.51–3.41)	0.576		
Alanine transaminase *	0.94 (0.41–2.15)	0.941		
Creatin phosphokinase *	2.06 (1.01–4.20)	**0.047**		
Lactate dehydrogenase *	5.13 (0.93–28.2)	0.060		
C-Reactive Protein *	1.33 (0.73–2.44)	0.357		
Ferritin *	1.00 (0.56–1.80)	0.999		
Total lymphocytes *	1.13 (0.54–2.36)	0.747		
D-Dimer *	1.23 (0.68–2.22)	0.501		
Fibrinogen *	0.57 (0.06–5.51)	0.629		
Interleukin-6 *	1.28 (0.63–2.58)	0.491		
sST2 (cut-off > 58.9 ng/mL ^†^)	6.32 (1.70–23.5)	**0.006**	9.73 (2.12–44.8)	**0.030**

BMI: Body Mass Index; CPK: Creatin Phosphokinase; DBP: Diastolic blood pressure; SBP: Systolic blood pressure. List of candidate variables included in multivariable logistic regression model (backward selection procedure): Age, Gender, BMI, previous history of diabetes, previous history of dyslipidemia, Estimated PAFI, creatin phosphokinase, Lactate dehydrogenase. AUC of the model: 0.818 (CI 95% [0.70–0.92]; *p* ≤ 0.001). ^†^ Point of highest sensibility and specificity (Sensitivity = 76·9%; Specificity = 62·1%; AUC = 0.693; *p* = 0.023). * Variables have been transformed using logarithmic polynomials. **Bold results** are statistically significant.

## Data Availability

Data supporting the findings of this study are available from the corresponding author upon reasonable request.

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
