# Peer review of "Early Measurement of Blood sST2 Is a Good Predictor of Death and Poor Outcomes in Patients Admitted for COVID-19 Infection"

_jcm, 2021, doi:10.3390/jcm10163534_

Round 1

Reviewer 1 Report

The manuscript reports that sST2 is a useful biomarker in identifying patients at risk of severe complications or death in COVID-19 infection. The authors showed that serum sST2 concentration on admission correlates with HOF prescription, transfer to ICU, length of hospital stay, and death, and is an independent predictor of worse prognosis. Although increased serum sST2 levels in COVID-19 infections have already been reported, the authors' study on sST2 may be useful in determining the direction of treatment for COVID-19 patients. There are a few issues that, if addressed, would increase the impact and strength of the manuscript.   1. Since sST2 is a decoy receptor for IL-33, it is important to clarify the relationship between sST2 and IL-33 in COVID-19 infection. Therefore, the authors should examine the serum IL-33 levels in healthy donors and COVID-19 patients. Although the authors mention in the discussion on lines from 307 to 309 that there are technical difficulties in measuring IL-33 levels, increased IL-33 in patient sera has been reported in asthma, NASH, SLE, etc., and can be measured by ELISA and Luminex assay. 2. In the discussion, the authors mention the increased levels of sST2, but little about the function of sST2. In particular, the authors should mention the protective effect of sST2 on lung inflammation induced by IL-33, along with the results of serum IL-33 measured in COVID-19 patients. 3. There are a few places in the text and figures where it is shown as sST-2. Please correct sST-2 to sST2 in the list shown below. Lines 194, 209, 284, and 349
Table 2, Figure 2, and Figure 3

Author Response

We are very thankful to the Reviewer for constructive comments that helped us improve the manuscript.

  1. Since sST2 is a decoy receptor for IL-33, it is important to clarify the relationship between sST2 and IL-33 in COVID-19 infection. Therefore, the authors should examine the serum IL-33 levels in healthy donors and COVID-19 patients. Although the authors mention in the discussion on lines from 307 to 309 that there are technical difficulties in measuring IL-33 levels, increased IL-33 in patient sera has been reported in asthma, NASH, SLE, etc., and can be measured by ELISA and Luminex assay.

We agree with the Reviewer that, given the observed differences in sST2 levels, it would have been very interesting to examine serum IL-33 levels in our patients to gain further understanding on the role of this cytokine in COVID-19 physiopathology. However, we decided not to invest our limited time and resources in measuring IL-33, because studying the role of IL-33/ST2 axis in COVID-19 was out of the scope of our study.

Our crowdfunded study was designed to explore the usefulness in evaluating COVID-19 prognosis of some tests (ultrasound, biomarkers) we use routinely in our studies on congestive acute heart failure, a condition with significant lung involvement. While lung ultrasound worked beautifully (Rubio-Gracia et al. Point-of-care lung ultrasound assessment for risk stratification and therapy guiding in COVID-19 patients. A prospective non-interventional study. Eur Respir J 2021, 2004283. doi:10.1183/13993003.04283-2020), most biomarkers yielded negative or inconclusive results. However, in our hands, sST2 clearly outperformed other prognostic markers being advocated at the beginning of the pandemic. Such result was in line with those of other groups being published in preprint form at the time. We believe this bears significant clinical value by itself. Exploring its physiopathology meaning would require much more work because sST2 or IL-33 levels may also be related to the cardiovascular responses to SARS-CoV2.

There is also a reason why we focused on biomarkers we were already very experienced with. While it is true there are several options in the market to determine IL-33, setting up a new technique involves investing some time and thorough testing, and besides time and funding constraints, we had very limited access to clinical samples.

Co-authors on this manuscript (LML, BGT) were involved in a thorough study about immunity in a larger cohort of COVID-19 patients. Of course, we would like to expand our collaboration looking into the role of IL-33/ST2 and to further validate sST2 prognosis value, but all this is pending on earning additional funding.

  1. In the discussion, the authors mention the increased levels of sST2, but little about the function of sST2. In particular, the authors should mention the protective effect of sST2 on lung inflammation induced by IL-33, along with the results of serum IL-33 measured in COVID-19 patients.

The Reviewer raises a very interesting point. However, we did not want to go deeper into the discussion of sST2 role in lung disease because it is out of the scope of our study and to avoid being too speculative. High serum sST2 most likely reflects induction by increased tissue (lung, but also other organs) IL-33 release, and thus works as a surrogate marker. Perhaps the decoy mechanism is overwhelmed, because higher serum sST2 correlates with worse prognosis, is clearly not protective. It is a very complex picture and that is why we refer to work by others presenting more in-depth hypothesis for the role of IL-33/ST2 axis in COVID-19 physiopathology (lines 301-324). Notwithstanding, we have revised the discussion to clarify this point. We have also updated references determining sST2 and IL-33 in serum/plasma from COVID-10 patients (lines 258-289). The available evidence suggests blood IL-33 changes might not be as consistent and bear less predictive value than those of sST2.

  1. There are a few places in the text and figures where it is shown as sST-2. Please correct sST-2 to sST2 in the list shown below. Lines 194, 209, 284, and 349.Table 2, Figure 2, and Figure 3.

We thank the Reviewer for pointing out this mistake we had overlooked. We changed all appearances to sST2 as suggested.

Reviewer 2 Report

General.

Based on the urgent need for early clinical predictor of Covid-19, the authors addressed the significance of IL-33/ST2 axis. They claim that the serum sST2 concentration on admission is an independent predictor of the disease prognosis. Although this parameter implicates some clinical value and prompts further investigation, the data presented in this manuscript have not supported its absolute significance.

The main result of this study is quite misleading due to inappropriate statistical analysis. In this kind of exploratory observation study, cohort analysis does not apply because no exposure/intervention discriminates the study population and the control. Experienced biostatisticians should advise to employ the appropriate methods and the data must be reevaluated.

Therefore, this manuscript is not acceptable, at least in its present form.

Specific.

Table 1. The variables should be compared between the outcome-positive and the outcome-negative groups, instead of P<25, 25<P<75 and P<75. If there is any parameter meets statistical significance, it may be a prognostic factor.

Table 2. Their chi-square test is fundamentally inappropriate, because the authors omitted one half of the patients (73 in 25<P<75 out of total 144). Comparing only P<25 and P>75 is intentionally misleading.

Figure 3. Log-rank test shows a clear discrimination between the above cutoff/below cutoff populations. However, the small difference between (0.95 vs. 0.75) may not be meaningful in the actual clinical settings, in addition to the fact that sST2 measurement is not available as a readily accessible laboratory test.

Minor.

Some grammatical and typographical points to be considered.

Line 39. “Regardless its broad phenotypic spectrum”: consider more appropriate expression.

Line 49. “miocarditis”: “myocarditis» ?

Line 63. “HF”: Should be “heart failure (HF)”, instead of explaining the abbreviation in Line 328.

Author Response

ANSWERS TO REVIEWER 2

We thank the Reviewer for the comments on our manuscript. We are sorry if we did not make a good job explaining the aims and the design of our study and appreciate the opportunity to clarify these issues.

We did not address the significance of IL-33/ST2 axis in our study, but were seeking biomarkers with prognostic value in COVID-19, which were and still are in much need. We focused on biomarkers we have experience with through our studies in the field of congestive acute heart failure and applied similar statistical analysis we have been employing for years in our group, to study prognostic value of different biomarkers. Our study is thus not interventional, and there is no control group. We present sST2 values in samples from healthy donors just to illustrate the timeline of sST2 elevation and, especially, that on discharge it has not returned to values observed in healthy subjects, which is a significant and possibly clinically relevant observation. Still, this sort of studies are similar to a prospective, longitudinal cohort study and thus similar analysis are appropriate to establish associations between outcomes and prognostic markers. In our study the control group would assimilate those patients not suffering the outcome.

In order to better present our results, we moved the description and analysis of serum sST2 levels in COVID-19 patients to the beginning of the Results section. Next, we present the clinical characteristics of our cohort, and analyze differences depending on sST2 distribution to show correlation between this marker and other clinical indexes of disease severity. Finally, we demonstrate sST2 levels at admission are independently associated with the primary adverse outcome and exhibit better predictive performance than other laboratory tests.

Specific 1: Table 1. The variables should be compared between the outcome-positive and the outcome-negative groups, instead of P<25, 25<P<75 and P<75. If there is any parameter meets statistical significance, it may be a prognostic factor.

The Reviewer raises a valid point. We have added Supplemental Table S1 comparing clinical and laboratory variables in the two sub cohorts. Here it can be appreciated sST2 levels were significantly higher in patients suffering the primary outcome. On the other hand, biomolecules previously considered as prognostic markers (e.g., IL-6, CRP, LDH, …) did not significantly differ. While this is an interesting initial observation, the next steps were to infer prognostic value and rule out confounding factors by running survival analysis, which is shown in Figures 2 and 3, and Table 3. Two additional variables, BMI and dyslipidemia, have been added to the logistic regression model shown in Table 3. These variables have been previously associated to worse prognosis and show differences in the comparison reported in Supplemental Table S1. However, only sST2 stood as independent marker of worse outcome.

Specific 2: Table 2. Their chi-square test is fundamentally inappropriate, because the authors omitted one half of the patients (73 in 25<P<75 out of total 144). Comparing only P<25 and P>75 is intentionally misleading.

At this point we would like to clarify the method we used to carry out the statistical analysis shown in Table 2. We used the ‘Linear to Linear Association’ test in SPPS, under the Chi-Square tests. This method is also known as Mantel-Haenszel test of trend (also Cochran-Mantel-Haenszel test) and is appropriate for the analysis of stratified categorical data. It can be used to test the association between an ordinal predictor or treatment and a binary outcome, while taking into account the stratification of the predictor variable.

In our case the ordinal predictor is the serum sST2 tercile at admission and the binary outcome is the incidence of primary or secondary outcomes (control being in either case the absence of outcome). Thus, the test provides significance for the comparison taking into account the stratification of the predictor, and of course includes variable values for all patients in the cohort, not only those in the extreme terciles. Stratification of patients according to terciles is a common practice in biomarker studies. The result of the test demonstrates there is significant association between being in a given sST2 tercile and suffering or not the primary outcome.

For comparison, we provide Supplemental Table S2 where we repeat the analysis considering the predictor (admission serum sST2) as a binary variable (below or above sensitivity threshold cut-off, calculated from ROC curve). It can be seen that significance is reduced because the stratification effect is not included. Notwithstanding, the association of sST2 with primary endpoint is maintained in this analysis (p=0.002).

We have added additional explanation under statistical methods and apologize for causing the misunderstanding that data could have been deliberately excluded from analysis.

Specific 3: Figure 3. Log-rank test shows a clear discrimination between the above cutoff/below cutoff populations. However, the small difference between (0.95 vs. 0.75) may not be meaningful in the actual clinical settings, in addition to the fact that sST2 measurement is not available as a readily accessible laboratory test.

We respectfully disagree with the Reviewer. Our analysis shows we can safely classify COVID-19 patients using the sST2 cutoff: those below the cutoff are unlikely to present the outcome. It is true there will be false positives in the group of patients above cutoff, but in the actual health situation focus should lay on identifying those in need of closer clinical follow up. In our cohort sST2 was the most sensitive biomarker at this task.

While it is true that measurement of soluble ST2 is not currently routine laboratory test in most places, it is very stable and easy to determine biomarker. A point of care kit to determine sST2 in the context of heart failure reached the market. There is a large body of literature advocating for sST2 evaluation in different conditions, especially lung and heart disease, but also cancer and kidney disorders. Demonstration of pathophysiological and prognostic value in COVID-19 could be the last evidence needed to convince health care providers that it should be added to routine available tests.

Minor Comments.

We thank the Reviewer for suggesting and pointing out grammar mistakes. These points have been corrected and are marked in the redlined version.

Round 2

Reviewer 1 Report

Unfortunately, the author's own data on IL-33 was not reflected in the revised manuscript, but instead the authors cited previous reports on IL-33 and showed the kinetics of IL-33 in COVID-19 in the discussion. The authors also improved the discussion on sST2.

The previous misspellings of sST2 have been improved, but there are some misspellings as ST-2 in the added Supplementary table 2. These need to be corrected.

Author Response

We are very thankful to the Reviewer for the comments that helped us improve the manuscript.

The previous misspellings of sST2 have been improved, but there are some misspellings as ST-2 in the added Supplementary table 2. These need to be corrected.

We have corrected these misspellings in the Supplementary table 2.

Reviewer 2 Report

In the revised manuscript, the authors added more detailed and precise description in Materials/Methods and Figure legends. These revisions significantly clarified their methodology, and the readers would now be able to evaluate the Results more appropriately. This work may prompt further investigations on sST2 in Covid-19 patients, which will eventually determine whether the biomarker is valuable in the actual clinical settings.

The reviewer considers that the manuscript can be published with minor grammatical and typographical refinement.

Line 281: What is ‘ARSD’?

Lines 323 and 338: What is ‘ADHF’? There is little meaning to abbreviate ‘heart failure’ as ‘HF’.

Author Response

We thank the Reviewer for the comments on our manuscript.

The reviewer considers that the manuscript can be published with minor grammatical and typographical refinement. Line 281: What is ‘ARSD’?. Lines 323 and 338: What is ‘ADHF’? There is little meaning to abbreviate ‘heart failure’ as ‘HF’.

We apologize for typographical and minor grammatical mistakes. We have corrected them.